# Principal Component Analysis for Very Heavy-Tailed Data

## Abstract

Principal component analysis (PCA) is a ubiquitous tool for dimensionality reduction and exploratory data analysis. However, most theoretical and empirical studies implicitly assume that noise is light-tailed. When data are corrupted by heavy-tailed noise, as is increasingly common (e.g. in omics or brain connectivity data), standard PCA techniques can fail dramatically. While recent work in robust statistics has addressed this problem in certain contexts, many existing methods remain sensitive to extreme outliers, performing poorly under truly heavy-tailed distributions. Furthermore, many of the methods which have been designed for heavy-tailed distributions do not scale well to large data sizes. In this work, we propose a novel algorithm for PCA that is designed for extremely heavy-tailed noise and which is computable for even very large data matrices. Our approach is designed to reduce sensitivity to such deviations while recovering informative low-rank structure. In the case of very heavy-tailed data with a large number of observations, we demonstrate significant improvements over classical PCA and existing robust PCA variants.

## 1 Introduction

Principal Component Analysis (PCA) is a fundamental technique in data analysis, extensively used for dimensionality reduction, data visualization, and feature extraction across a variety of scientific and engineering domains (Fan et al., 2018). Standard PCA, including modern variants as appearing in Ledoit & Wolf (2020), however, is fundamentally based on eigendecomposition of the empirical covariance matrix, implicitly relying on light-tailed assumptions regarding noise distribution. Under such conditions, PCA exhibits desirable theoretical properties, including consistency and robustness to moderate perturbations (Benaych-Georges & Nadakuditi, 2012). Nonetheless, real data increasingly challenge these assumptions, particularly in high-dimensional settings like genomics and finance, where heavy-tailed distributions are common (Fan et al., 2021; Ke et al., 2019).

Heavy-tailed data are characterized by extreme 'outlier' deviations from the mean, which severely affect the eigendecomposition of the sample covariance matrix (Soshnikov, 2002; Auffinger & Tang, 2016). Consequently, classical PCA methods tend to produce highly inaccurate estimates of principal directions, as these are disproportionately influenced by outliers or extreme values. This phenomenon severely undermines PCA's applicability to modern datasets, necessitating robust methodologies explicitly designed for heavy-tailed noise.

In recent years there have been a small number of PCA methods designed to deal with heavy-tailed noise; see for instance Ke et al. (2019). These have mostly been validated (either theoretically or empirically) for specific classes of distributions or under conditions such as finite fourth moments which, despite being much more flexible than standard PCA contexts, is nonetheless restrictive relative to the context we consider here. Our interest is in extremely heavy-tailed noise, allowing (for instance) infinite variance.

In the following section, we introduce a novel PCA algorithm. It is motivated by an empirical observation specific to heavy-tailed data, namely that population-level principal components are distributed over low-dimensional subspaces spanned by sample principal components. The framework of our method is based on repeated subsampling and projections onto principal component subspaces, exploiting the consistent geometric alignment of principal directions within these subspaces.

The geometric strategy is exceedingly simple, but we find that it is sufficient to improve on the accuracy of other methods. Furthermore, our method depends on three hyperparameters $N, P, R$, but we demonstrate that the performance of our algorithm is largely insensitive to these choices, the only key being that $R$ is sufficiently large.

We validate our method on synthetic heavy-tailed data sets, demonstrating improved performance for extremely heavy-tailed data, such as under infinite variance. We also show that our method shows an improved ability to find self-consistent signal directions in real-world transcriptomic and synaptic connectivity data sets. Although our method is motivated by the problem of PCA for heavy-tailed data, it also works well in the light-tailed case. The only situation in which we have found our method to suffer relative to alternatives is when the number of observations is small.

Although not part of the design, our algorithm is well adapted to large data sizes. Other methods either depend on convex optimization over spaces of matrices, which become very slow for large matrix sizes, or depend on a number of computations which grows prohibitively with the number of observations. Our method uses only standard linear-algebraic computations and (in terms of computation time) is insensitive to the number of observations. Unlike the other available methods, ours appears to be both readily computable and accurate for data sets with thousands of features or observations.

## 2 Heavy-tailed principal component analysis

The celebrated "BBP phase transition" discovered by Baik et al. (2005) and extended by Benaych-Georges & Nadakuditi (2012) says that, for data generated by certain (light-tailed) random matrix models, the leading eigenvectors of the sample covariance matrix are at a *deterministic angle* from the leading eigenvectors of the population covariance matrix. This applies in the theoretical asymptotic limit of an infinite number of observations and infinitely many observations, although it can be empirically seen to a high level of accuracy for reasonably large matrices (for instance, $100 \times 500$). According to this asymptotic theory, PCA based directly on the sample covariance matrix will have a predictable defect as an estimator of the principal components of the population-level covariance.

This can be contrasted with the case of heavy-tailed distributions, which presents different phenomena, as established initially in Soshnikov (2002) and taken to a higher level of generality in Auffinger & Tang (2016). In the simplest case of a heavy-tailed 'pure noise' matrix $Z$ (i.e. one with i.i.d. entries following a heavy-tailed distribution), the leading eigenvectors of the sample covariance $ZZ^{\mathsf{T}}$ are, with high probability, close to coordinate vectors. The coordinates in question are governed by the loci of the most extreme 'outliers' in $Z$, i.e. the largest entries in absolute value.

We note here, somewhat informally, that this extends to more realistic random matrix models than those generated as pure noise. Consider the information-plus-noise model $X = P + Z$, where $P$ is a 'signal matrix' and $Z$ is a 'noise matrix' as in the previous paragraph. Under the heavy-tailed assumption, $Z$ will contain extremely large entries, so that the matrix norm of $Z$ will be very large relative to that of $P$. As such, on the scale of $P$, the data matrix $X$ can be treated as a small perturbation of $Z$, so that the leading eigenvectors of the sample covariance $XX^{\mathsf{T}}$ will be small perturbations of the leading eigenvectors of $ZZ^{\mathsf{T}}$. (See Kato (1995) for the relevant perturbation theory.) That is, they will be approximately coordinate vectors and hence approximately orthogonal to true population-level principal components.

Our interest in this paper is the classical PCA problem of estimating the leading eigenvectors of the population covariance matrix. According to the BBP phenomena and related work, the sample covariance matrix is of limited utility in the light-tailed case. According to the previous paragraph, it is essentially useless in the heavy-tailed case. This highlights the need for novel PCA algorithms in the heavy-tailed case. We present our algorithm below, and briefly review some other algorithms from the literature in the subsequent section.

### 2.1 Our algorithm

In the light-tailed case, the BBP phenomena asserts that for certain data matrix models, the leading eigenvector $u$ of the population covariance matrix is nontrivially (and predictably) correlated with the leading eigenvector $v_1$ of the sample covariance matrix. Our algorithm is motivated by the

Table 1: Projection of the population-level principal component $u$ onto sample eigenvectors $v_1, v_2, \ldots$

| | Light-tailed model | | | Heavy-tailed model | |
|---|---|---|---|---|---|
| $i$ | $(u \cdot v_i)^2$ | $\sum_{j=1}^{i}(u \cdot v_i)^2$ | $i$ | $(u \cdot v_i)^2$ | $\sum_{j=1}^{i}(u \cdot v_i)^2$ |
| 1 | 0.9986 | 0.9986 | 1 | 0.0157 | 0.0157 |
| 2 | 0.0001 | 0.9986 | 2 | 0.0599 | 0.0757 |
| 3 | 0.0000 | 0.9986 | 3 | 0.2166 | 0.2923 |
| 4 | 0.0000 | 0.9986 | 4 | 0.0015 | 0.2938 |
| 5 | 0.0000 | 0.9986 | 5 | 0.1006 | 0.3944 |
| 6 | 0.0000 | 0.9986 | 6 | 0.5203 | 0.9146 |
| 7 | 0.0001 | 0.9987 | 7 | 0.0264 | 0.9410 |
| 8 | 0.0000 | 0.9987 | 8 | 0.0006 | 0.9416 |

empirical observation that, in the heavy-tailed case, $u$ is instead nontrivially correlated with the first several eigenvectors $v_1, v_2, \ldots$ of the sample covariance. This is illustrated in Table 1, where we consider the data model

$$X = \begin{pmatrix} u & u_2 & \cdots & u_{100} \end{pmatrix} \operatorname{diag}(20, 1, \ldots, 1)V + \frac{1}{\sqrt{200}}Z$$

which is an instance of the information-plus-noise model detailed later in our section on synthetic experiments. Here $u, u_2, \ldots, u_{100}$ are random orthonormal vectors in $\mathbb{R}^{100}$ and $V$ is a random orthogonal $200 \times 200$ matrix, while the noise matrix $Z \in \mathbb{R}^{100 \times 200}$ has i.i.d. entries. In the 'light-tailed' part of Table 1, the entries of $Z$ are drawn from the Student distribution with 100 degrees of freedom (df), while in the 'heavy-tailed' part we use the Student distribution with 1.5 df.

Here, for heavy-tailed noise, it is clear that the population-level principal component $u$ is not well-approximated by any of the leading sample eigenvectors $v_1, v_2, \ldots$, but that a reasonably good approximation can be found inside the span of the first eight eigenvectors. This is very unlike the BBP phenomena found for light-tailed noise, where the leading eigenvector of the population covariance has some (in this case, quite large) correlation with the leading sample eigenvector and random (small) correlation with all of the others.

Based on this observation, we hypothesize the existence of a positive integer $P$ such that the population-level principal component $u$ is likely to be contained in the span of the leading $P$ eigenvectors of the sample covariance. By subsampling the data matrix $X \in \mathbb{R}^{p \times n}$ to $p \times N$ submatrices, it is possible to generate many $P$-dimensional linear subspaces of $\mathbb{R}^p$ which are likely to (nearly) contain $u$. We identify our estimate of $u$ as the unit vector which comes the closest to being contained in each of these subspaces.

This can be quantified as follows. Given a linear subspace $\Pi$ of $\mathbb{R}^p$ with orthonormal basis $v_1, \ldots, v_P$, the orthogonal projection of $u \in \mathbb{R}^p$ onto $P$ is $\sum_{i=1}^{P}(v_i \cdot u)v_i$, and so the (Euclidean) distance from $u$ to $\Pi$ is $\|u - \sum_{i=1}^{P}(v_i \cdot u)v_i\|$. Given linear subspaces $\Pi^1, \ldots, \Pi^R$, with $v_1^\alpha, \ldots, v_P^\alpha$ an orthonormal basis of $\Pi^\alpha$, our quantitative interpretation of "the vector which is the closest to being contained in each $\Pi^\alpha$" is the minimizer of

$$L(u) = \sum_{\alpha=1}^{R} \left\| u - \sum_{i=1}^{P}(v_i^\alpha \cdot u)v_i^\alpha \right\|^2$$

over unit vectors $u \in \mathbb{R}^p$. This can be reformulated by viewing each $v_1^\alpha, \ldots, v_P^\alpha$ as an orthogonal matrix $V^\alpha \in \mathbb{R}^{P \times p}$ (i.e. $V^\alpha(V^\alpha)^{\mathrm{T}} = \mathrm{I}_{P \times P}$). Now $(V^\alpha)^{\mathrm{T}}V^\alpha u \in \mathbb{R}^p$ is the vector $\sum_{i=1}^{P}(v_i^\alpha \cdot u)v_i^\alpha$, so the above "loss function" can be rewritten as

$$L(u) = \sum_{\alpha=1}^{R} \|u - (V^\alpha)^{\mathrm{T}}V^\alpha u\|^2 \equiv \sum_{\alpha=1}^{R} (u - (V^\alpha)^{\mathrm{T}}V^\alpha u)^{\mathrm{T}}(u - (V^\alpha)^{\mathrm{T}}V^\alpha u).$$

or equivalently as

$$L(u) = R\|u\|^2 - \sum_{\alpha=1}^{R} u^{\mathrm{T}}(V^\alpha)^{\mathrm{T}}V^\alpha u.$$

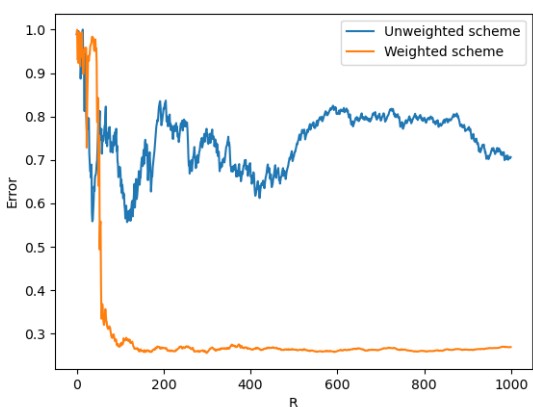

Figure 1: Use of weighted selection. If we were to use uniform weighting to select our submatrices $\tilde{X}$, our PCA estimate would depend in an unstable way on the choice of $R$. Using a non-uniform weighting stabilizes the estimate and allows us to simply choose $R \gg 1$. See also Figure 2(a) for further illustrations of stability.

Subject to the unit-vector constraint on $u$, the first term can be ignored, and minimization of the above function is equivalent to finding the leading eigenvector of

$$\sum_{\alpha=1}^{R} (V^\alpha)^{\mathrm{T}} V^\alpha,$$

according to the well-known variational interpretation of the first eigenvalue and eigenvector of a symmetric matrix.

We summarize our algorithm as follows:

> **Hyperparameters:** positive integers $N$, $R$, $P$ with $N < n$ and $P < p$
> **Given:** data matrix $X \in \mathbb{R}^{p \times n}$
> Set $W = 0 \in \mathbb{R}^{p \times p}$. Repeat $R$ times:
> > Randomly choose $N$ of the columns of $X$ to get $\tilde{X} \in \mathbb{R}^{p \times N}$
> > Compute the $P$ leading (orthonormal) eigenvectors of $\tilde{X}\tilde{X}^{\mathrm{T}} \in \mathbb{R}^{p \times p}_{\mathrm{sym}}$
> > Arrange them into a matrix $V \in \mathbb{R}^{P \times p}$ and add $V^{\mathrm{T}} V$ to $W$.
> The leading eigenvector of $W$ is the estimate of the leading principal component.

However, the methodology of the random choice of columns is also important. Our selection of $N$ columns is not done uniformly at random, but is instead based on a weighted distribution so that columns with large Euclidean norm (i.e. those with outlying entries) are less likely to be selected. This makes it unlikely for outlying entries to consistently enter into the subselected $\tilde{X}$, making the computed $V$ minimally distorted by outliers. Our choice of weight is so as to be proportional to the inverse of the Euclidean norm of a column. As shown in Figure 1, the use of non-uniform selection is key to the success of our algorithm. Without it, the hyperparameter choice discussed in the following section would be significantly more complicated.

Our algorithm can be viewed as a Monte Carlo method. This is especially convenient to state in the case $P = N$. One is given data points $x_1, \ldots, x_n \in \mathbb{R}^p$. Consider the random linear map $L : \mathbb{R}^p \to \mathbb{R}^p$ given by orthogonal projection onto the span of the randomly-drawn subset $x_{i_1}, \ldots, x_{i_N}$ of data points. Our PCA estimate is a Monte Carlo estimate of the leading eigenvector of the expectation $\mathrm{E}[L]$. No matter how heavy-tailed the distribution generating the data points may be, this is a reasonable Monte Carlo problem, since the random linear map $L$ is strongly bounded, valued as it is in the compact space of orthogonal projections. As such, this perspective sheds some partial light on the robustness of our method. Moreover, it is strongly reflective of our parameter choice $R \gg 1$, as $R$ represents the number of Monte Carlo iterates.

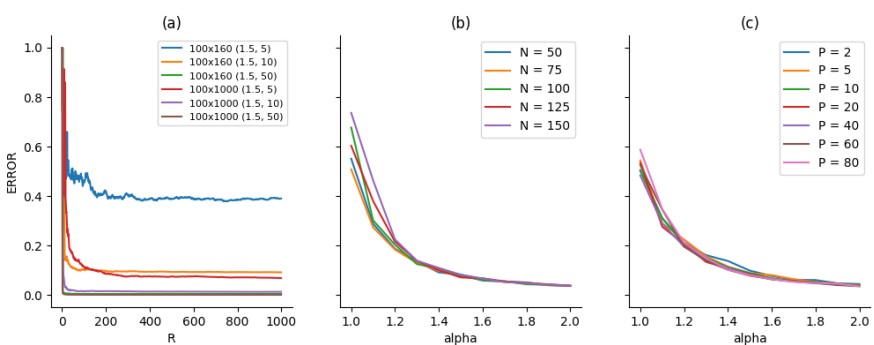

Figure 2: Effect of parameter choices on performance

## 2.2 PARAMETER SELECTION AND COMPUTABILITY

In our algorithm, there are three hyperparameters $P, R, N$ that need to be chosen. However, in practice we find that, subject to minor constraints, the dependence of performance on these parameters is not very significant. Figure 2(a) shows the result of changing $R$; each curve corresponds to a fixed data matrix, with $P = N = 50$. Since each curve levels out quickly, optimal and stable performance is achieved by taking $R$ suitably large. In practice, we usually take $R = 500$ or $R = 1000$. As noted above, this is why our *non-uniform* selection is key. Figure 2(b) shows the effect of changing $N$. We enforce the constraint $N \geq P$ since a $P$-dimensional principal subspace has to be computed for a $p \times N$ matrix, so that $N < P$ could only introduce randomness into the subspace. Although the effect is not terribly strong, it appears that increasing $N$ has a slightly negative effect on performance, so in practice we choose $N = P$. Figure 2(c) shows the minimal effect of changing $P$. Here, each point along each curve corresponds to the 20-fold average performance of our algorithm with $R = 500$ and $N = P$. It is visually apparent that changing $P$ has only a negligible effect on performance, and not with any clear trend. However, if $P$ is selected too large (such as $P = p$), performance will be significantly degraded. See the Supplement for similar plots with different parameter values.

In addition to the above motivation, the choice $N = P$ strongly aids the computability of our algorithm. If $N > P$, it is necessary to compute ($R$ many times) an orthogonal projection onto the span of the $P$ leading eigenvectors of a $p \times N$ matrix. When $N = P$, this is equivalent to computing ($R$ times) the orthogonal projection onto the column space of a $p \times N$ matrix. Thus is it unnecessary to compute eigendecompositions and the QR decomposition suffices.

Unless specified otherwise, our default choice of parameters will be $P = N = \frac{p}{2}$ and $R = 1000$.

## 2.3 RELATED WORK

Several approaches have been proposed to address the challenges of PCA and covariance estimation posed by heavy-tailed data. The "linear eigenvalue shrinkage" estimator from Ledoit & Wolf (2004) is perhaps the best known covariance matrix estimator. More recently, Ledoit & Wolf (2020) have identified a *nonlinear* eigenvalue shrinkage estimator which is proved to be optimal in various situations. We emphasize here that, optimality as covariance matrix estimators notwithstanding, the matrices estimated by a "shrinkage" approach such as Ledoit–Wolf's or Nadakuditi (2014) have the strong constraint of having the same eigenvectors as the sample covariance matrix. For this reason, they are not of any greater (direct) use for PCA than the sample covariance matrix.

Beginning with Maronna (1976) and Tyler (1987), there have been many works in the robust statistics literature which deal with estimation of the covariance matrix for specifically heavy-tailed data. The most common context is that of *elliptically contoured distributions*, which are affine-linear transformations of spherically symmetric distributions that may be either light- or heavy-tailed. Many of these works deal with a small number of features, such as bivariate data, but big-data problems have also been considered; see Goes et al. (2020) for recent work and further references, and Ke et al. (2019) for a survey. Han & Liu (2018) introduced Elliptical Component Analysis (ECA),

in which the data matrix is used to compute the multivariate Kendall's tau statistic. In the case of infinite data sampled from elliptically contoured distributions, it is shown that the eigenvectors of the tau statistic coincide with the population-level PCA directions.

Candès et al. (2011) and Chandrasekaran et al. (2011) studied "Robust PCA" algorithms based on decomposing a data matrix as the sum of a sparse matrix and a low-rank matrix, principal component estimates being computed via the latter matrix. Although not apparently designed with heavy-tailedness in mind, these algorithms appear to be relevant since a heavy-tailed noise matrix is approximately sparse when viewed on the scale of its largest entries.

Modeled on Huber's *median of means estimator* from univariate robust statistics, Minsker (2015) estimates the covariance matrix as the *geometric median* of the sample covariance matrices of a partition of the data matrix into subsamples. There are some schematic similarities between this algorithm and our own, principally in being divide-and-conquer algorithms based on subsampling the data. However, Minsker's subsampling amounts to a single random partition of the observations into some set number of groups. By contrast, we perform (in principle) arbitrarily many subsamples, each chosen randomly by a *preferential* (non-uniform) selection which appears to be key to the success of our algorithm. More broadly, the fundamental difference between these algorithms is that Minsker's algorithm is a convex optimization based on the sample covariance matrices of data subsamples while ours is direct linear algebra based only on leading principal eigenspaces.

We briefly mention some other related works which cannot be compared directly to ours. Roy et al. (2024) considers a "robust PCA" algorithm which is based on fitting an elliptically contoured distribution to data. However, the key functional degree of freedom in such a distribution is assumed to be known in advance, making this unsuitable as a general PCA algorithm. Since the robust PCA of He et al. (2023) is in fact a kernel PCA based on the characteristic function (Fourier) transform, it can be used for dimension reduction but not for the problem of estimating principal components. Lastly, Mohammadi et al. (2015) and Mohammadi (2022) fit multivariate stable distributions to data. This is naturally viewed as the heavy-tailed generalization of fitting multivariate Gaussians to data; see Nikias & Shao (1995) or Samorodnitsky & Taqqu (1994) for information on the stable distributions. However (unlike the special case of Gaussians) the parameters of multivariate stable distributions possess infinitely many degrees of freedom and the interpretation of principal components therefrom does not appear straightforward.

Lastly, we note that Lemma 6 of Fan et al. (2019) is essentially the same as the "quantification" paragraph of the previous section. However, the problem addressed by that paper is distinct. There, different observations of a data matrix are stored in different locations, and the problem is to "aggregate" the PCA estimates of each such data submatrix into a single PCA estimate (otherwise unavailable) for the whole matrix. By contrast, we actively break apart the data matrix and then aggregate PCA estimates in order to obtain an *improved* PCA estimate. Moreover, heavy-tailedness does not appear to have any role in their work.

## 3 SYNTHETIC EXPERIMENTS

### 3.1 GENERATIVE MODEL

Our data-generating model is often called the **information-plus-noise model** in the literature. Here we consider the case of a unidimensional signal and noise drawn from a Student distribution; the parameters are the number of features ($p$), the number of observations ($n$), the degrees of freedom of the Student distribution ($\alpha$), and the signal strength ($\kappa$). The $p \times n$ data matrix is

$$X = UDV + n^{-1/2}Z$$

where $U \in \mathbb{R}^{p \times p}$ and $V \in \mathbb{R}^{n \times n}$ are random orthogonal matrices drawn from the Haar (i.e. uniform) distribution on the orthogonal group, $Z \in \mathbb{R}^{p \times n}$ has $pn$ i.i.d. entries drawn from the Student distribution with $\alpha$ degrees of freedom, and $D \in \mathbb{R}^{p \times n}$ is a matrix with $D_{ij} = 0$ if $i \neq j$ and $D_{ii} = 1$ if $i \geq 2$ and $D_{11} = \kappa$. In this model (assuming always $\kappa > 1$), the first column of $U$ is the population-level principal component and is the object of estimation. We estimate this vector, $u$, by unit vectors $\hat{u}$, and measure the error by $1 - (u \cdot \hat{u})^2$. Zero error corresponds to $\hat{u} = \pm u$.

We note that $\alpha$ governs the heavy-tailedness of our data model, with smaller values of $\alpha$ corresponding to heavier-tailed distributions. When $\alpha \geq 2$, the data-generating model has finite variance

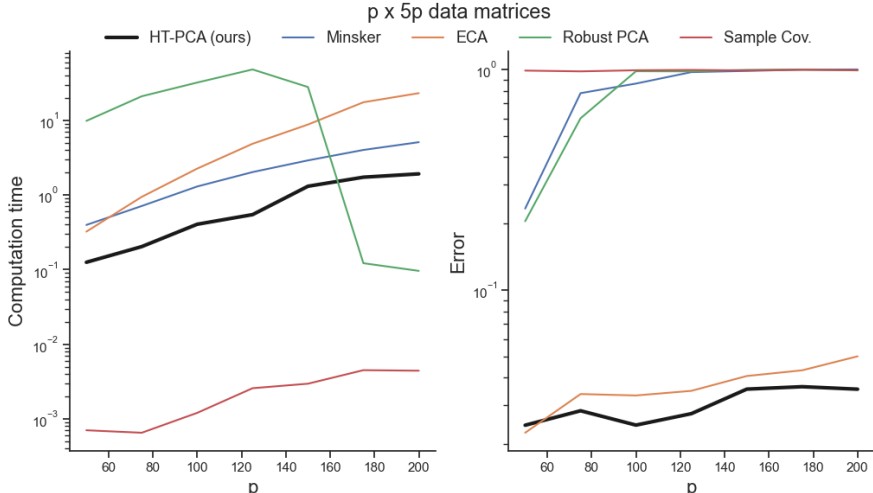

Figure 3: The left figure shows the 5-fold average computation time for data matrices of varying sizes with $n = 5p$. The right figure shows the corresponding error in the PCA estimate. Here $\alpha = 1.5$ and the signal strength is 10. Note the decrease in Robust PCA computation time maps to a severe increase in error. Standard PCA based on the sample covariance matrix is extremely fast but inaccurate.

and the population-level covariance matrix is $UDD^{\mathrm{T}}U^{\mathrm{T}}$, from which it is clear that $u$ is indeed the population-level principal component. When $\alpha < 2$, the model has infinite variance and the population-level covariance matrix is undefined. However, even in this case, $u$ represents the directional structure in the model (just as in the $\alpha \geq 2$ case), so we still refer to it as a population-level principal component.

### 3.2 COMPARISON ALGORITHMS AND COMPUTATION TIME

We compare the performance of our algorithm against the following alternatives: standard PCA using the sample covariance matrix, the elliptical component analysis (ECA) of Han & Liu (2018), Robust PCA (RPCA) from Candès et al. (2011), and the geometric median method from Minsker (2015).

As described above, standard PCA is not useful for heavy-tailed data; we include it here only to establish an absolute baseline. ECA is simple to implement and requires no hyperparameter selection, but it is costly to run when there are a large number of observations. Minsker's algorithm, in its version using the *thresholded geometric median*, depends on a hyperparameter $0 \leq \nu \leq 1$ as well as the selection of $k$, denoting how many groups to partition the observations into. In our numerical experiments, we have selected $\nu = 0.5$ and $k = 10$ in accordance with the experiments in Minsker's paper. Minsker's algorithm and Robust PCA both rely on convex optimization, which becomes computationally expensive for large data sizes. For the latter we use the implementation available at `https://github.com/dganguli/robust-pca`. Computation times are reported in Figure 3, which demonstrates that ECA and our algorithm achieve competitive errors, but that ours is faster by an order of magnitude. The discrepancy increases for larger $p$. For example, in the case of a $500 \times 2500$ matrix, ECA takes around 15 minutes to run while ours takes around 10 seconds. For much larger matrices, such as the transcriptomic and synaptic connectivity data sets appearing in Section 4, ECA appears to be completely impractical.

We also compare against the simple-minded but practical method of using standard PCA after deleting some set number of 'corrupted' observations from the data matrix. Here, we delete the 50 observations of the highest Euclidean norm. This is an arbitrary choice, but it is noteworthy that in many cases it already improves upon other methods. The problem of adaptively choosing how many observations to delete is an interesting and natural problem which, to the best of our knowledge, has

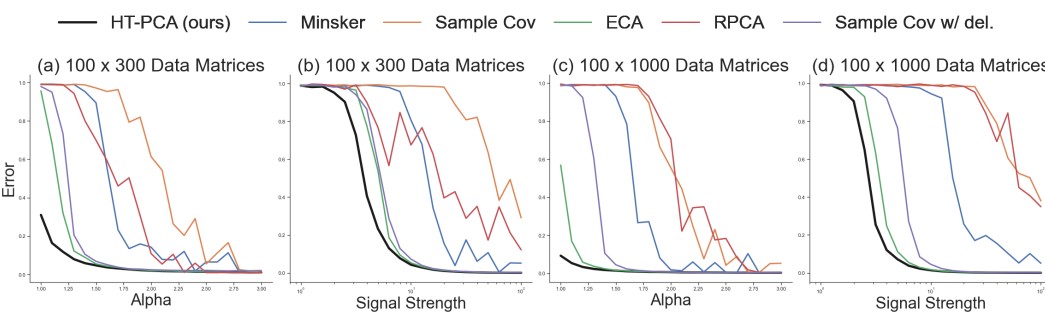

Figure 4: Comparison of PCA algorithms on synthetic data. Each point on a curve represents an average performance, with fixed parameters, over 20 runs. In **(a)** and **(c)** we fix the signal strength $\kappa$ at 10 and vary $\alpha$ from 1 to 3; in **(b)** and **(d)** we fix $\alpha$ at 1.5 and vary the signal strength from 1 to 100.

not been considered in the literature. We note that this method is as fast to compute as the "Sample Cov." method as reported in Figure 3.

### 3.3 NUMERICAL SIMULATION

In Figure 4 we show the parameter dependence of all six algorithms in question, using the parameters detailed above. For our algorithm we use $P = N = 50$ and $R = 1000$. We note especially the poor performance of Robust PCA, in some cases being indistinguishable from direct use of the sample covariance matrix.

As expected, increasing $\alpha$ (making the data more light-tailed) has a positive effect on the performance of each algorithm. We note in particular the markedly superior performance of our algorithm in the extremely heavy-tailed case of $\alpha \approx 1$. (We note that $\alpha > 1$ is the cutoff point where the data generating model has finite expectation.) In subfigures (b) and (d), each curve necessarily starts at an error of 1, since a signal strength of 1 corresponds to an absent signal. The key behavior is how quickly each curve departs from this value as the signal strength is increased.

It is notable that increasing $n$ from 300 to 1000, which amounts to the inclusion of extra data, actually has a deleterious effect on the performance of RPCA, while it does not appear to help the performance of Minsker's algorithm. By contrast, as should be expected, the inclusion of additional data does improve the performance of our algorithm and ECA.

## 4 TRANSCRIPTOMIC AND SYNAPTIC CONNECTIVITY APPLICATIONS

### 4.1 DROSOPHILA TRANSCRIPTOMIC DATA SET

RNA sequencing data has the distinguishing and challenging characteristics of both high sparsity and heavy tails (Fan et al., 2021). Although a complete analysis remains elusive, since sparsity is not one of the considerations of our algorithm, we find signs of improvements when using our algorithm for analysis.

We use a data set consisting of single-cell transcriptomes of developing and adult olfactory receptor neurons in Drosophila, from McLaughlin et al. (2021). This contains count data of 17,807 genes across 1,371 cells, interpreted as 1,371 observations in a 17,807-dimensional feature space. To (partially) address the effects of sparsity, we remove all genes from the data set which are nontrivially expressed in fewer than 100 cells, thereby removing 12,252 genes and leaving a $5555 \times 1371$ matrix.

Since we lack ground-truth principal components for this data, we measure the self-consistency of a PCA algorithm by *split-half reproducibility*. Here we randomly divide the 1,371 observations into one group of 685 and a complementary group of 686, forming submatrices $X \in \mathbb{R}^{5555 \times 685}$ and $X' \in \mathbb{R}^{5555 \times 686}$ of the data set constructed in the previous paragraph. The leading principal components $v_1, v_1'$ are computed (as unit vectors), and their consistency is measured by $1 - (v_1 \cdot v_1')^2$. If the PCA algorithm in question is self-consistent, this should be close to zero. If instead it is close

to one, it is evident that the result of the PCA algorithm depends strongly on which observations happen to be part of the data set.

We found it to be beneficial to additionally preprocess the data by normalizing each gene by mean and standard deviation, so that we are effectively doing PCA with the sample correlation matrix. Without this additional step, the split-half reproducibility error is extremely small, but with found eigenvectors which are very close to coordinate vectors. We speculate that this could be related to the sparsity of the data matrix, or to the nonnegativity of its entries, both of which pose problems for interpretation and which do not presently have any counterpart in our synthetic models. This is a clear question for future research.

With this normalization, across 20 runs, the average error of our algorithm with parameter choice $P = N = 30$ and $R = 500$ is **0.105**, with principal components not close to coordinate vectors (the typical largest-magnitude entries thereof being around $0.07$ or $0.08$). We interpret this as a sign that our algorithm is finding a self-consistent and nontrivial signal.

### 4.2 MICRONS SYNAPTIC CONNECTIVITY DATA

Neuronal connectivity in the brain is often characterized by heavy-tailed degree and weight distributions, with a small number of highly connected "hub" cells or strong synapses dominating network structure. These characteristics violate assumptions of light-tailed or Gaussian-distributed inputs that underlie classical dimensionality reduction techniques like standard PCA. To better capture dominant low-dimensional structure in such data, we apply our heavy-tailed PCA. In particular, we evaluate our HT-PCA on a synaptic connectivity matrix: a directed sparse matrix of $50,594$ neurons and approximately 2.6 million nonzero connection weights (see Supplement for details).

To identify the most non-Gaussian and heavy-tailed subnetworks, we estimated an $\alpha$-stable tail exponent for each row of the matrix, fitting to the largest 5% of outgoing weights using maximum likelihood (using scipy.stats.levy_stable.fit). Among rows with $\geq 100$ nonzero entries, we selected the $1,500$ neurons with lowest estimated $\alpha$ and extracted the $1,500 \times 1,500$ induced submatrix used for all downstream analysis. The resulting distribution of $\alpha$ values confirmed the presence of heavy-tailed structure ($\alpha$: mean = 0.97, median = 0.13). This extremely non-Gaussian submatrix serves as a highly challenging benchmark for our comparison of our algorithm and Minsker's method. By randomly splitting the columns of the $1,500 \times 1,500$ synaptic connectivity matrix to yield $1,500 \times 750$ heavy-tailed submatrices (10 times, using the same random indices for each comaprison of our algorithm with Minsker's; see Supplement), our algorithm (parameters $P = N = 50$; $R = 100$) showed significantly higher split-half reproducibility (mean cosine similarity = 0.569, SD = 0.304) than Minsker's geometric median estimator, which produced near-zero cosine similarity (mean = 0.0049, SD = 0.0055), indicating that our algorithm was better able to discover conserved patterns of projection (or "globally conserved projection motifs") in this sample.

## 5 CONCLUSION AND LIMITATIONS

We presented a new model-free PCA algorithm tailored to very heavy-tailed data, where classical and existing robust methods often fail. By aggregating principal subspaces from weighted subsamples, our approach avoids covariance estimation, scales efficiently, and remains robust even when variance is infinite. Across synthetic benchmarks and challenging biological datasets, it consistently outperforms other comparable estimators, while showing minimal sensitivity to hyperparameters. These results show our method to be a practical step forward in robust, scalable dimensionality reduction for modern high-dimensional, heavy-tailed data.

Still, our study is primarily empirical. While we provide extensive evidence that our estimator performs well under very heavy-tailed noise, a theoretical analysis explaining the phenomena in Table 1, or the parameter dependence in Figure 2, remains open. Further in cases with a small number of observations, our algorithm performs strikingly worse than the alternatives (see Supplement). Interestingly, because our method builds on standard PCA as an intermediate step, there is a natural opportunity to combine it with other robust PCA techniques for potentially greater accuracy. Exploring such hybrid approaches and expanding formal guarantees are important future directions.

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

# A APPENDIX

## A.1 OPEN ACCESS TO DATA AND CODE

All data in the paper was generated or is publicly available (The Drosophila data may be found here: `https://pubmed.ncbi.nlm.nih.gov/33555999/`; the MICrONS data can be loaded using the code we provide using the CAVEclient Python library, and are at `https://www.nature. com/articles/s41586-024-07765-7`). We make our code to reproduce the above results available as open source under the MIT license on GitHub: [link will appear here; for now see code provided in ZIP file].

## A.2 HARDWARE REQUIREMENTS

All experiments were conducted on MacBook Pro laptops equipped with 16GB of RAM. Training times varied depending on model complexity and dataset size, with smaller experiments completing within minutes and the largest experiments requiring a few hours. The computational setup demonstrates that our proposed methods are accessible and do not require specialized high-performance computing infrastructure, even on larger data, making them readily reproducible by the broader research community.

## A.3 LACK OF SENSITIVITY TO MINSKER'S PARAMETER

In our simulations, we have found that Minsker's algorithm is not very sensitive to the parameter $\nu$. This is illustrated in Figure 5, which replicates the context of Figure 4(a) of the paper for varying choices of $\nu$. (However, here, for simplicity, we report only single outcomes, not 20-fold averages.) For this reason, we consistently take $\nu = 0.5$ in our simulations.

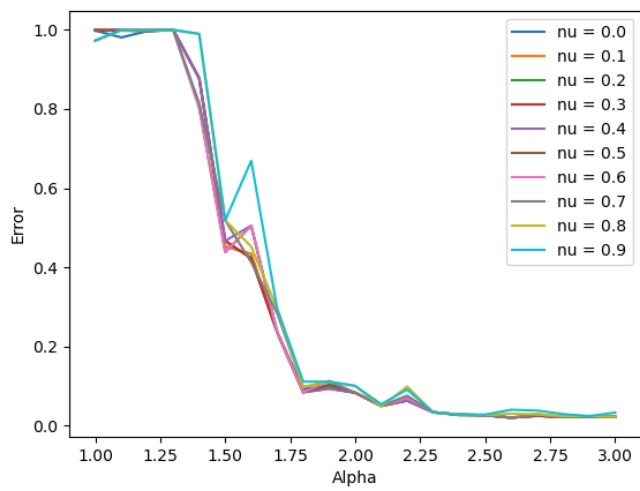

Figure 5: Dependence of Minsker's algorithm on parameter $\nu$

### A.4 FURTHER PLOTS ON PARAMETER DEPENDENCE

Here we present further plots like those of Figure 2 of the main text. Figure 6 shows multiple plots like those of Figure 2(a). Each of the nine plots shows the results of five simulations with exactly the same parameters. It can be seen that the dependence on $R$ is highly stable, and that an optimal error is reached well before our choice of $R = 500$.

Figure 7 shows the dependence of our algorithm on the choice of $P$, extending Figure 2(c) from the case $100 \times 160$ to the matrix sizes $100 \times 400$, $200 \times 400$, and $400 \times 800$. Here we additionally show standard deviations in our 10-fold averages. (Accordingly, for visual clarity, we here show fewer choices of $P$.) As can be seen from these plots, our algorithm is not sensitive to the choice of $P$, unless it is chosen as an extreme value. For example, when $P$ is chosen to be 100 for a $100 \times 400$ matrix, the leading $P$-dimensional principal subspace of any subselected data matrix is simply the entire space $\mathbb{R}^{100}$, and no information can be gleaned. In this case, the significant jump from $P = 90$ to $P = 100$, illustrated in Figure 7, should be noted.

### A.5 MICRONS DATA PREPROCESSING AND AND BENCHMARK CONSTRUCTION

We constructed our heavy-tailed synaptic benchmark from the publicly available MICrONS Phase 1 dataset ("minnie65_public") using the CAVEclient Python API (version 5.4.5). Following Elabbady et al. (2025), we restricted our analysis to neurons located within a cortical band spanning 18,000 to 25,000 nm in depth. Using the "proofreading_status_neurons" and "synapses_pni_2" tables, we filtered for neurons classified as excitatory or inhibitory and included only synaptic edges where both the pre- and post-synaptic partners were located within the specified depth band and labeled as neurons. This yielded a sparse, directed synaptic connectivity matrix containing 50,594 neurons and approximately 2.6 million nonzero connection weights.

To identify a challenging, heavy-tailed subnetwork for evaluation, we estimated an $\alpha$-stable tail exponent for the outgoing weights of each neuron. For each row with at least 100 nonzero entries, we fit a symmetric $\alpha$-stable distribution to the top 5% of outgoing weights using the scipy.stats.levy_stable.fit Python function, fixing location and scale parameters (floc=0, fscale=1). The resulting distribution of $\alpha$ values was heavy-tailed, with mean $\alpha = 0.97$ and median $\alpha = 0.13$, indicating the presence of extreme non-Gaussian structure. From this distribution, to make computation reasonable we selected the 1,500 neurons with the lowest estimated $\alpha$ values and extracted the induced $1,500 \times 1,500$ submatrix for all downstream evaluations.

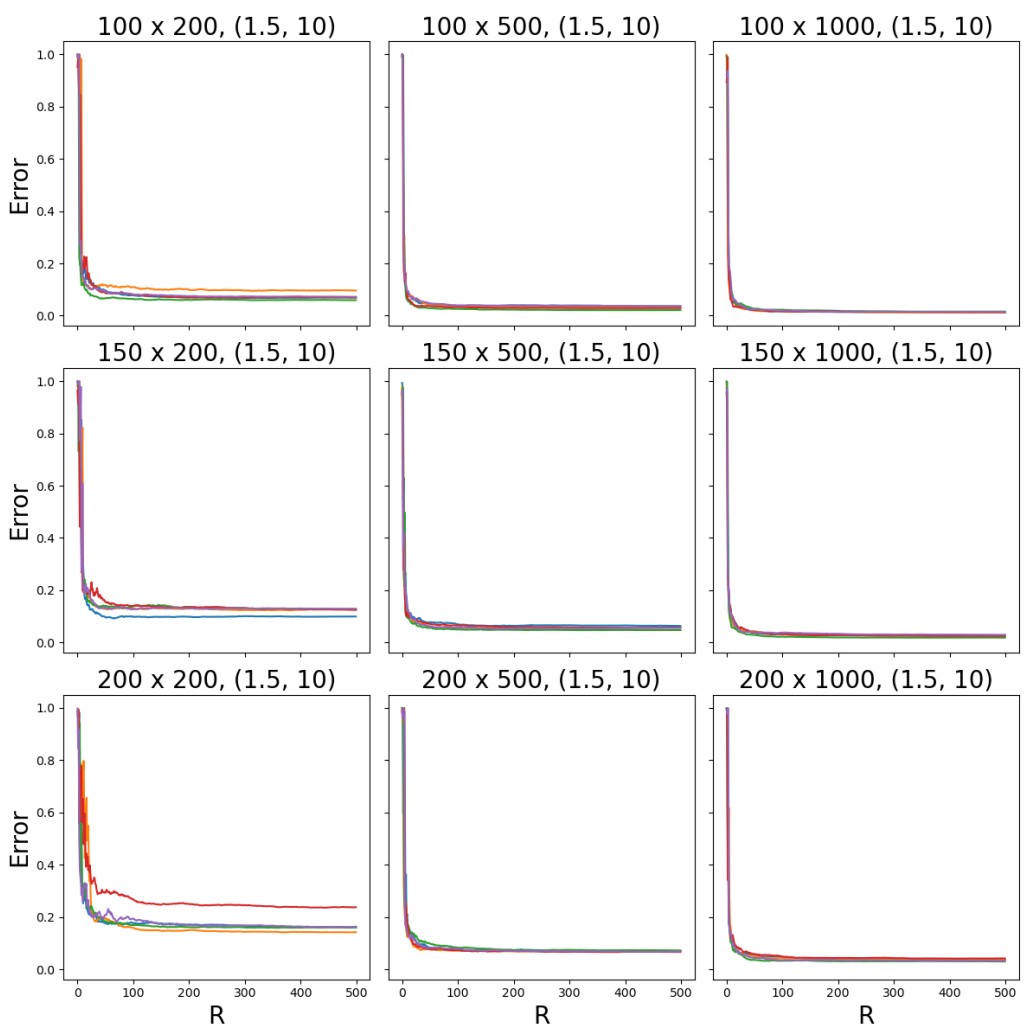

Figure 6: Dependence on $R$, cf. Figure 3(a).

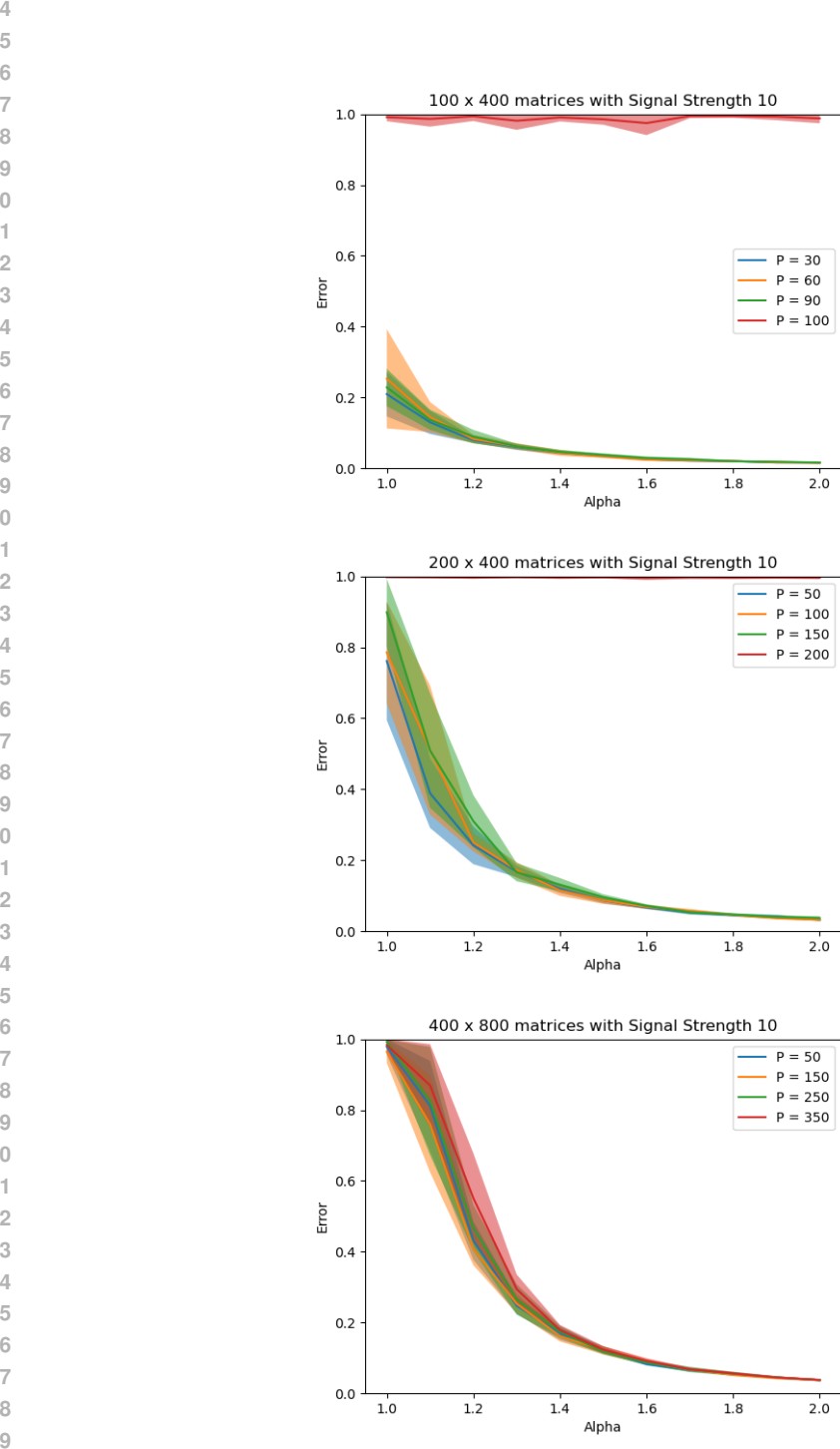

Figure 7: Dependence on $P$, cf. Figure 3(c). These plots show 10-fold averages together with standard deviations.

To assess reproducibility, we generated 10 independent split-half samples by randomly partitioning the columns of this submatrix into two equal-sized sets of 750 columns each. All comparisons between our HT-PCA method and Minsker's geometric median estimator used the same random column partitions for each trial. We then computed cosine similarity between subspaces recovered on each half-sample to evaluate consistency. For HT-PCA, we used subsample size $P = N = 50$ and number of subsamples $R = 100$. Minsker's method was implemented using the cvxpy convex optimization package in Python, with parameters $\nu = 0.5$ and $k = 3$. All analyses were conducted using Python 3.10, with NumPy, SciPy, and scikit-learn, and took under 1 hour to run. Additional implementation details and code are available in the accompanying code.

### A.6 FURTHER PLOTS ON ALGORITHM COMPARISON

Here we include further plots like Figure 4 of the main text, for a greater range of parameter values. As there, we show 20-fold averages; here we additionally represent the standard deviation (truncated to remain between 0 and 1) via the shaded regions. Figure 8 shows the results for various matrix sizes and signal strength 10; Figure 9 shows various matrix sizes and $\alpha = 1.5$. For our algorithm, the parameter values are $R = 1000$ and $P = N = \lfloor \frac{p}{2} \rfloor$; for Minsker's algorithm, the parameter values are $k = 10$ and $\nu = 0.5$.

### A.7 LICENSES FOR EXISTING ASSETS

All models and code are original or are clearly imported and indicated as such (as open source Python packages). All data in the paper were generated or are publicly available. We will make our code available as open source under the MIT license.

### A.8 BROADER IMPACT

This work develops theoretical and algorithmic advances for principal component analysis under heavy-tailed distributions, a fundamental problem in multivariate statistics and machine learning. Heavy-tailed data naturally arises in many domains including finance (asset returns, risk modeling), network analysis (degree distributions, traffic patterns), climate science (extreme weather events), and genomics (gene expression with outliers). Our robust PCA method could improve analysis in these areas, potentially leading to better risk assessment, more reliable dimensionality reduction for downstream tasks, and improved handling of datasets with natural outliers or measurement errors. While the contributions are primarily methodological, the generality of our approach means it could be applied to sensitive domains involving personal data. Although PCA itself is often used for privacy-preserving dimensionality reduction, practitioners should remain mindful of privacy implications when applying any dimensionality reduction technique to personal or sensitive datasets.

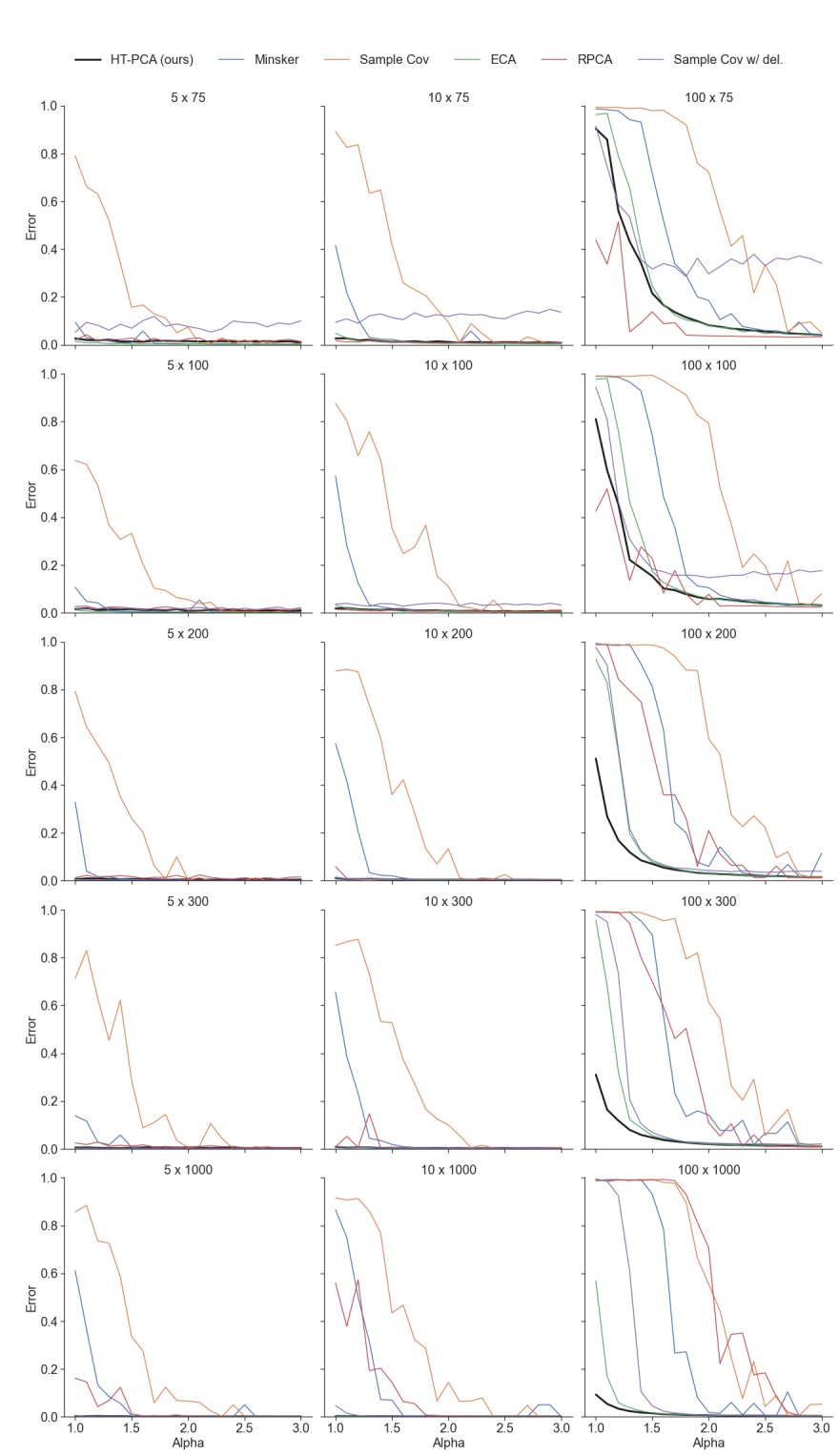

Figure 8: Dependence on $\alpha$. All plots share a common $x$-axis and $y$-axis. Each title shows $p \times n$.

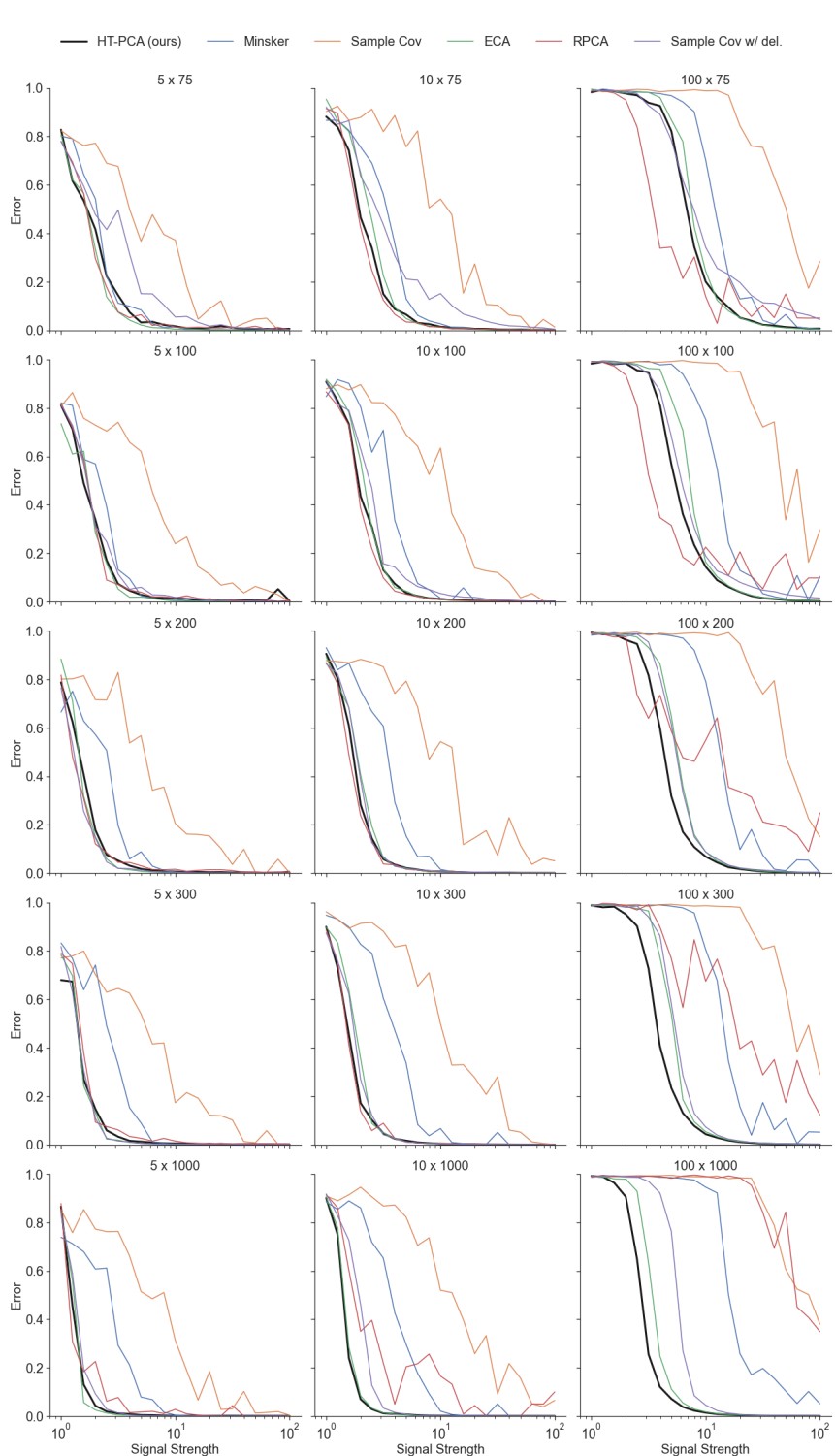

Figure 9: Dependence on signal strength. All plots share a common $x$-axis and $y$-axis. Each title shows $p \times n$.

