# OpenReview forum: "Principal component analysis for very heavy-tailed data"
_ICLR.cc/2026/Conference — Submitted to ICLR 2026_

### Official Review · Reviewer_dhEA · 2025-11-01

**Soundness:** 1
**Presentation:** 2
**Contribution:** 2
**Rating:** 2
**Confidence:** 2

**Summary:**

This paper studies the problem of performing principal component analysis (PCA) under extremely heavy-tailed noise, where classical PCA and many existing robust PCA approaches fail. The authors observe that for heavy-tailed data, the true principal component may lie in the span of several leading sample eigenvectors rather than aligning with the top one. Based on this observation, they propose a heuristic algorithm that repeatedly subsamples columns of the data matrix, computes leading eigenvectors of each subsample, and aggregates these subspaces to recover the principal direction. The method incorporates a weighted sampling strategy to avoid repeatedly selecting outlier-dominated columns. Empirically, the approach demonstrates improved robustness over classical PCA and several robust PCA variants on synthetic data and two biological datasets. The paper claims scalability and practical advantages, especially in high-dimensional scenarios with heavy-tailed noise.

**Strengths:**

- The paper addresses an important and challenging problem: PCA for extremely heavy-tailed data, where standard methods are known to break down.
- The proposed method is simple, scalable, and easy to implement using basic linear algebra routines, making it potentially useful for large-scale applications.
- The authors provide extensive experimental evaluation on synthetic data and two real-world datasets, demonstrating that the approach can achieve improved empirical robustness compared to many baselines.
- The observation that the signal may lie in the span of multiple sample principal components under heavy-tailed noise is interesting and worth further theoretical investigation.

**Weaknesses:**

The primary concern is that the algorithm is entirely heuristic and lacks any theoretical guarantees. While heuristics are valuable in practice, the paper does not provide sufficient justification for the proposed approach beyond empirical observation. In particular:

- The method does not come with formal guarantees regarding recovery accuracy, convergence, or robustness, unlike prior work in robust PCA and heavy-tailed estimation.
- The motivation relies heavily on a qualitative empirical observation, but no theoretical explanation or analysis is offered to support the key claim.
- The experiments, although extensive, are not sufficiently diverse to fully establish the reliability of the heuristic. For such a method, more varied real-world benchmarks and stronger empirical gains are necessary to justify its contribution.
- In the absence of theory, the paper risks lacking generality; it remains unclear under which regimes or distributional assumptions the algorithm can be expected to perform well.

**Questions:**

- Can the authors provide theoretical insights, even partial, into why aggregating PCA subspaces from subsampled data approximates the true principal direction under heavy-tailed noise?
- How sensitive is the algorithm to the choice of hyperparameters in practice, particularly for datasets with different scales and tail behaviors?
- Would the method still hold up if evaluated on a broader set of real-world datasets beyond transcriptomics and neural connectivity data?

---

> ### Author Response · Authors · 2025-12-03
>
> We thank the reviewer for their comments and questions. We understand the perceived key weakness to be the lack of theory. Please see our response to Reviewer Jcbn for a theoretical analysis in the infinite-data limit which highlights how our weighted resampling procedure effectively creates light-tailed data, to which concentration inequalities can be applied.
>
> In response to the other concern about further comparisons, also see our response to Reviewer Jcbn, where we add comparisons to the truncation-based PCA from Ke et al. (2019) and spatial-sign PCA.
>
> The sensitivity of our algorithm to hyperparameters is analyzed in Section 2.2 of our paper and Figures 2, 6, and 7. As discussed there, we suggest default choices of $P=N$ for computational efficiency, $P=N=\frac{p}{2}$ for convenience, and (based on how quickly the plots of Figures 2a and 6 level off) $R=1000$ as a conservative choice. For an additional analysis in the case of the Drosophila data set, see the response to Reviewer mB9s. There we find a slightly greater dependence on the choice of $P=N$; performance degrades slightly if these parameters are too large. However, our default parameter choice still provides good results for this data set. At the cost of computational efficiency, it would be straightforward to adaptively choose $P=N$ so as to minimize the (average) split-half reproducibility error.
>
> It would, of course, be highly interesting to analyze additional real-world data sets. However, we believe that our two real-world examples are sufficient to show that our algorithm performs well on challenging data which does not arise from simple synthetic models. Even these examples are not meant to be definitive; we believe that it is necessary to understand the role of data preprocessing, in addition to the role of sparsity. However, we consider this to be most appropriate to future research.

---

### Official Review · Reviewer_mB9s · 2025-11-04

**Soundness:** 3
**Presentation:** 3
**Contribution:** 2
**Rating:** 4
**Confidence:** 3

**Summary:**

This paper introduces a simple and scalable method for performing principal component analysis when data exhibit heavy-tailed noise, a setting where classical PCA and many robust variants often fail because the sample covariance is dominated by a few extreme observations. The proposed approach repeatedly draws random subsamples of the data, computes the top-$P$ principal components for each subsample, and aggregates the resulting subspaces by averaging their projection matrices. The final estimate of the leading principal direction is obtained as the top eigenvector of this aggregated projection matrix, which stabilizes the estimate by diluting the influence of heavy-tailed outliers across many subsamples. The method uses only standard linear algebra operations and empirically outperforms classical PCA, geometric-median PCA, and convex robust PCA on synthetic and biological datasets, particularly in regimes with infinite-variance noise.

**Strengths:**

1.  Introduces a simple yet effective geometric aggregation approach for PCA under heavy-tailed noise, based on repeated subsampling and subspace averaging. The method is conceptually simple yet addresses a challenging regime (including infinite-variance data) rarely handled by existing algorithms.

2. Demonstrates substantial gains over classical PCA, Minsker’s geometric-median PCA, and convex robust PCA across synthetic and real-world datasets, particularly under extreme heavy-tailed noise.

3. Achieves scalability by relying only on standard linear-algebraic primitives (SVD, eigen-decomposition), avoiding convex optimization and achieving near-linear time in data size.

4. The algorithm is transparent and easy to implement, offering intuitive insight into why subspace aggregation stabilizes principal directions.

5. Includes empirical sensitivity analyses over hyperparameters $(P, R, N)$ and tests on diverse biological datasets (transcriptomic and synaptic connectivity), highlighting robustness and generality.

6. Effectively connects the BBP phase transition and random-matrix theory results to the degradation of PCA under heavy-tailed noise, grounding the algorithm’s rationale.

**Weaknesses:**

The paper lacks formal theoretical guarantees, deeper analysis of hyperparameter sensitivity, and a comprehensive comparison to established robust-scatter PCA frameworks (e.g., Tyler's (1987), ROBPCA (Hubert et al. 2012) etc.). These omissions limit the perceived depth of contribution. Here are some of the key weaknesses want to outline:

1. The method is supported primarily by geometric intuition and empirical evidence, but lacks formal theoretical guarantees. In particular, the paper does not provide asymptotic analysis, finite-sample error bounds, perturbation-theoretic results, or convergence and robustness guarantees.

2. The algorithm’s dependence on $(N, P, R)$ is discussed qualitatively, but tuning strategies are heuristic and not systematically analyzed, particularly for real-world data.

3. The relationship to established robust-scatter and shape-based PCA approaches (e.g., Tyler's M-estimator, ROBPCA, spatial-sign PCA) is underdeveloped, limiting clarity on conceptual novelty.

4. Baselines focus on convex robust PCA and geometric-median PCA, omitting newer high-dimensional or probabilistic heavy-tail estimators (e.g., Catoni-type covariance, truncation-based PCA, Lerman \& Maunu 2018).

5. While the method performs well across the presented experiments, the paper offers little guidance on when the proposed approach may fail or be inappropriate. In particular, there is no analysis of how performance depends on the data regime (e.g., high-dimensional settings with $p\gg n$, low sample size scenarios, or weak signal-to-noise conditions), nor any diagnostic tools for practitioners to assess whether HT-PCA is likely to provide a reliable estimate on a given dataset. Given that robustness methods can degrade sharply outside their intended regimes, a clearer discussion of failure cases, limitations, and practical checks (e.g., subsample stability tests) would strengthen the utility and transparency of the proposed approach.

**Questions:**

In relation to the weaknesses outlined  above, here are the questions for the authors:

1. Instead of taking the mean of projection matrices, have you considered alternative aggregation measures (e.g., geometric or median subspace averaging)?

2. Can you provide any theoretical intuition or formal result about convergence of the leading eigenvector of $W$?

3. How should practitioners choose the hyperparameters $(N,P,R)$ in different regimes?

4. Could your approach be extended to recover multiple principal components simultaneously?

---

> ### Author Response · Authors · 2025-12-03
>
> We thank the reviewer for their time and clear questions.
>
> Weaknesses:
> 1. We understand your primary concerns to be the lack of theory and the need for more comparisons. For some theoretical analysis based on concentration inequalities, see our response to Reviewer Jcbn.
>
> 2. We dispute somewhat the characterization of our analysis of hyperparameter dependence. The dependence on $N,P,R$ is discussed quantitatively for synthetic data in Section 2.2 and Figures 2, 6, and 7. There we find a minimal dependence on $N=P$ (a constraint we always make for the sake of computational efficiency), prompting our default choice of $P=N=\frac{p}{2}$ and, based on how quickly the plots in Figure 2 and 6a level off, $R=1000$. We note here that this choice contains a typo and should be corrected to $P=N=\frac{1}{2}\min(p,n)$ to allow for the `few data' case $n<p$.
>
>     In the case of the Drosophila data set, we have run additional tests and do not find a strong dependence on our parameter. We maintain the choice $R=1000$ but vary the choice of $P=N$. Over 20-fold runs, we find the following average (split-half reproducibility) errors:
>
>     | $P=N$ | Error |
>     |-------|----------------|
>     | 5     | $0.117 \pm 0.014$ |
>     | 50    | $0.102 \pm 0.010$ |
>     | 100   | $0.102 \pm 0.016$ |
>     | 200   | $0.111 \pm 0.018$ |
>     | 250   | $0.117 \pm 0.020$ |
>     | 300   | $0.129 \pm 0.038$ |
>     | 342   | $0.142 \pm 0.028$ |
>
>     We note that the final choice corresponds to our aforementioned default choice. Here, unlike in our synthetic experiments, there is a visible decrease in performance quality as $P=N$ becomes larger. However, the effect is small and our default choice (the last row) still performs well. We note that it would be straightforward to adaptively choose $P=N$ so as to minimize this split-half reproducibility error. However, this would sacrifice the computational efficiency of our algorithm.
>
> 3. Tyler's M-estimator defines the estimated covariance matrix as the solution of a certain algebraic equation defined by the data points. Spatial-sign PCA applies standard PCA after replacing each data point by its projection to the unit sphere in $\mathbb{R}^p$. We are not aware of any conceptual similarity of these algorithms to ours.
>
>     The ROBPCA estimator is based on applying a robust covariance estimator to the subspace spanned by selection of a certain (somewhat arbitrary) number of `least outlying' data points. The selection of non-outlying data points suggests a small resemblance to our algorithm. However, unlike ROBPCA, our selection is non-deterministic and repeated multiple times, with the PCA results from different repetitions aggregated to form our overall estimate.
> 4. For additional comparisons, see our response to Reviewer Jcbn, where we report new results comparing to the truncation-based PCA from Ke et al. (2019) and spatial-sign PCA.
> 5. Although our algorithm generally works well, we have found that when $n$ is small, it (along with the other estimators in question) is notably outperformed by the truncation-based estimator of Ke et al. This is consistent with our response to your question on theory, as it is based on the infinite-data limit.
>
> Questions:
>
> 1. Subspace averaging methods (as in Marrinan et al. (CVPR 2014), ``Finding the subspace mean or median to fit your need'') are not directly applicable to our method. Our method is based on selecting a vector which is most common to a collection of subspaces, as opposed to averaging together several subspaces (which would not result in a distinguished direction, as needed for PCA).
>
>     Following your question, we have considered the entrywise median of the projection matrices (instead of the mean). However, it does not perform as well. For example, with $(p,n)=(100,500)$ and with our customary choice of $(R,P,N)=(1000,50,50)$, the 20-fold average error using the median is $0.45\pm 0.29$ as opposed to $0.18\pm 0.03$ using our approach.
>
> 2. This seems to duplicate Weakness \#1; see our response there.
> 3. This seems to duplicate Weakness \#2; see our response there.
> 4. Our approach does indeed directly extend to recovering multiple eigenvectors. In our derivation in Section 2.1, we restricted attention to the leading principal component for the sake of simplicity. However, it extends without major modification to the case of larger-dimensional eigenspaces; the result is that one instead looks for the leading eigenvectors of $W$, as opposed to only the single leading eigenvector. We can draw attention to this in Section 2.1 and give the more general derivation in the appendix.

---

### Official Review · Reviewer_ibbZ · 2025-11-05

**Soundness:** 3
**Presentation:** 3
**Contribution:** 3
**Rating:** 6
**Confidence:** 4

**Summary:**

This paper provides a different idea for what a principal component for data should be when dealing with am emphasis on heavy tailed data.

They suggest the following procedure for defining the principal component of data:
- Given a set of vectors $\vec x_1, \ldots, \vec x_n \in \mathbb R^p$
- Take a random subsample of those vectors, say $\vec{x}_{s_1}, \ldots, \vec{x}_{s_N}$ for $N \ll n$
- Let $\mathbf V \in \mathbb R^{p \times P}$ contain the top $P$ left singular vectors of $\vec{x}_{s_1}  \ldots  \vec{x}_{s_N}$
- The principal component is then the top eigenvector of $\mathbb E[VV^\top]$

This principal component is then estimate by simple monte carlo: Generate many such $V$ matrices, form an empirical estimation of $\tilde W = \mathbb E[VV^\top]$, and return the top eigenvector of $W$.

$V$ is not sampled uniformly at random. It uses importance sampling which is inversely proportional to the norms of the vectors.


The paper gives evidence suggesting that for heavy-tailed data, this recovers a more natural notion of a principal component when compared to classical PCA (i.e. returning the top eigenvector of the sample covariance matrix).

Evidence is empirical throughout the paper; theory is not provided.

**Strengths:**

I think this is a really interesting text. Overall, I'm inclined to accept, pending some adjustments to the text.

The intuition underlying the proposed estimator makes sense, and is a simple linear algebraic notion. I've never seen it used for this purpose, but I've certainly see people analyze the expected projection $\mathbb E[VV^\top]$ in theoretical problems (see eg Thm 3.1 of [this paper](https://arxiv.org/pdf/2208.09585); no need to cite this or anything just being thorough about the connection).

The paper provides convincing evidence that this notion of a principal component is meaningful, and that it is empirically useful.

The paper is well written and was even kinda fun to read!

I'm no expert on the statistical side of PCA, so I can't really comment effectively on the originality of this work relative to prior work. Taking their long prior work section at face value, this seems like a valuable contribution!

**Weaknesses:**

The paper suffers four core weaknesses, not all equal:
1. The experiments lack confidence intervals (plus other smaller issues on the figures)
2. The experiments fail to consistently and effectively compare the proposed PCA method to alternative PCA methods on
3. There is not a very crisp formalization of what makes a PCA method "good" for heavy tailed data
4. There is not a clear notion of how to produce more than one principal component


Let's start with experiments.

The figures in this paper are slightly disastrous.
- Despite the proposed method being a randomized algorithm, and the variance of randomized methods for PCA often having non-trivial confidence intervals, none of the experiments seem to have any confidence intervals. In my view, EVERY plot should contain confidence intervals for work like this. I personally prefer seeing the median error with 10/90 quantiles or 25/75 quantile; though mean +- standard deviation is okay. Line 242 says the code was run 20 times, so this should be an easy fix.
- Figure 2 has no real caption (page 5)
- Figures throughout the paper have unexplained parameters. The notion of "ERROR" and "alpha" isn't defined until after figure 2.
- Printed on paper, it's very hard to read the axes of many figures. The text should be larger on the axes (and in some legends)

Next, the paper lacks some baseline comparisons and confuses me at some points.
- "Sample Cov w/ Del." is absent from Fig 3 for some reason
- Section 4.1 (page 8) studies the "self-consistency" metric on real data, but only reports the error achieved by the proposed PCA method, and does not show the error achieved by the other methods considered on synthetic data. No confidence interval on the error is given.
- Section 4.2 (page 9) studies the same error metric on a different real data source, but only reports the error achieved by two PCA methods. The metric here used is confusing as the authors refer to both "self-consistency" and "mean cosine similarity" but only define the former, and perhaps only report the latter? Either way, the metric used here is confusing, and not enough estimators are compared.
- Section 4.2 has a high standard deviation of their error metric, nearly as large as the average value of the metric. Some further discussion about runtime to lower that standard deviation would be good (I know it's discussed elsewhere in the paper; but it needs acknowledgement here as well)


Next, let's get more conceptual. There's not a clear notion of what a good PCA method is.
The paper proposes two tests that a good PCA method should achieve on heavy-tailed data
1. Have good "self-consistency": if you split a dataset in half and run your PCA method on each half, it should return nearly identical vectors
2. Work on data from a specific generative model that has noise distributed as a heavy-tailed Student distribution

These are both... good things we want from PCA, but neither one really is a fundamental notion of what good PCA should be defined as. I'd like to see a more fundamental model for what the authors consider good PCA to be. The authors have this interesting ansatz that with heavy-tailed noise, the fundamental principal components should be distributed amongst the first few eigenvectors of the sample covariance matrix. I'd love to see this pushed a step further, into a potential guess for what a good formalization of PCA would be.

I'll acknowledge that my question here is somewhat underspecified; I'm not sure the authors have a good super formal notion of what PCA should be defined as, and I don't want to give them an undue burden to do such a thing. But if they have a more formal idea, I'd love to see that written out more. (to clarify, not an algorithm, but a more statistical notion of what a principal component should be)



This bleeds into my final topic -- the fact that this paper only considers producing a single principal component.
It's a very obvious question to ask: How should I generate a second principal component?
And how about the $k^{th}$?
Explicit iterative deflation may be needed as in LazySVD; or maybe just returning the top $k$ eigenvectors of the monte carlo projection suffices?
I think this should be acknowledged within this paper, at least to some minimal extent.

**Questions:**

## List of typos & recommended edits

_ Feel free to ignore anything in here you disagree with, without any need for further discussion _

1. [60] Specify what community these real-world datasets come from. Neuroscience?
2. [68] Last sentence here is phrased too strong. Maybe "Unless many other PCA methods for heavy-tailed data..."
3. [79] "reasonably small matrices"? at ICLR, a 100 by 500 matrix is small but already shows what you want it to show
4. [144] Usually, to me, Pi is a projection not a subspace
5. [throughout] P and p both being symbols in this paper is kinda annoying... maybe swap p for d, or swap P for k?
6. [154] Usually, to me, V is a tall matrix so that V'V is the identity and VV' is a projection. Transpose the definition?
7. [Fig 1] Specify the data used to generate table 1
8. [Throughout] Actually formalize the method used to importance sample. IID sampling with/without replacement? Wrt squared col L2 norms, or non-squared norms? Any smoothness/regularization used?
9. [189] I think this is just Courant-Fisher. Would be good to name-drop that here.
10. [Sec 2.3] This is very long. Shorten this a bunch. Will help you with the page limit.
11. [314] You're using kappa for both kurtosis and signal strength. Split these two different things up.

---

> ### Author Response · Authors · 2025-12-03
>
> We thank the reviewer for their very thorough review. We agree with the proposed edits and will implement them. (However, as regards Line 314, we are not aware of any reference to kurtosis in our paper.)
>
> 1. Thank you for calling our attention to the problems with our presentation of the figures. As you suggest, these will be easy to fix.
> 2.
>     * “Sample Cov w/ Del.” is absent from Figure 3 because the point is to compare the time efficiency of different algorithms, and for this purpose it is effectively identical to “Sample Cov.” We can make this explicit in the caption.
>     * We were not able to run the alternative methods on our real-world data sets because of issues with the time of computation. We alluded to this in lines 372-373, but we can make this more explicit. Among our newer alternatives, spatial-sign PCA performs well, achieving an average error of $0.098\pm 0.011.$ This is very slightly better than the performance of our algorithm ($0.102\pm 0.010$); as such, we only claim our algorithm as an alternative that performs well.
>     * Here we will fix our language for consistency. “Self-consistency” and “mean cosine similarity” refer to the same thing.
> 3. We do not have a comprehensive theory of what a good PCA algorithm should accomplish, or how success should be measured. (As you suggest, this would be a tall order.) For synthetic light-tailed data, it is generally accepted that PCA should recover the leading components of the population-level covariance matrix. For example, in the data model $X=AZ$ considered in our theory (see the response to Reviewer Jcbn), the population-level covariance $\text{E}[XX^\top]$ is (up to a scalar factor) $AA^\top$ under the natural assumptions of symmetry and independence for the entries of the (light-tailed) noise matrix $Z$. The population-level covariance $\text{E}[XX^\top]$ is no longer meaningful if $Z$ is drawn from a heavy-tailed distribution, but $AA^\top$ remains meaningful and its eigenstructure still reflects the directional structure of the data-generating process. As such, in the heavy-tailed context we find it well-motivated and meaningful to refer to the leading eigenvectors of $AA^\top$ as the population-level principal components, and it is then natural to measure the accuracy of a PCA algorithm by how well it recovers these vectors. The same considerations go through for the information-plus-noise model $X=A+Z/\sqrt{n}.$
>
>     In the case of real-world data, even in the light-tailed case the situation is not so clear. It is for this reason that we use the 'split-half reproducibility'. Since we do not have access to a ground-truth principal component, we instead measure the self-consistency of our algorithm by comparing the results after randomly dividing the observations into two groups. In and of itself, self-consistency in this sense does not say anything about PCA: all it says is that the algorithm in question is stably estimating some quantity. The connection is made by analyzing synthetic data, in which case that quantity can be concretely identified as a principal component.
> 4. You are correct to assume that further principal components are recoverable as the leading eigenvectors of the Monte Carlo projection. In Section 2.1, we restricted our attention to the leading principal component for simplicity, but we can call this to attention and include the detailed generalization of 145--190 in the appendix.

---

### Official Review · Reviewer_Jcbn · 2025-11-05

**Soundness:** 2
**Presentation:** 3
**Contribution:** 2
**Rating:** 2
**Confidence:** 5

**Summary:**

This paper proposes a method to compute the leading singular vector(s) of a matrix, motivated by scenarios where the matrix data is heavy-tailed (which is known to degrade the performance of other methods)

The method involves iterative taking random columns (ie datapoints) and finding the principal components of this, then adding up these found principal components and finally fining the principal component of the agglomerate.

There is no rigorous guarantee of when or if the method works, but it is evaluated on synthetic and real datasets.

**Strengths:**

Method has been clearly described.

**Weaknesses:**

The main weakness of this paper is that there is no rigorous guarantee that this method will work (or even, a characterization of specific scenarios of when it will work). Most existing methods for Robust PCA have such guarantees, and it is important to establish at least a basic rigorous guarantee of when such an algorithm will work (and, will work better than simple PCA)

Also, the paper is missing many works on robust PCA as comparison baselines, e.g.

https://arxiv.org/abs/1010.4237

https://arxiv.org/abs/2305.02544 (and references therein)

Etc.

**Questions:**

Not currently

---

> ### Author Response · Authors · 2025-12-03
>
> We thank the reviewer for their time in reading our paper.
>
> # Theory
> In our own view, the empirical aspect of our work is the most important. However, in response to your criticism (in addition to that of other reviewers) we have developed some theory for our algorithm. In the following, we consider the infinite-data limit for the data model $X=AZ$, with $A\in\mathbb{R}^{p\times p}$ and $Z\in\mathbb{R}^{p\times n}$, the latter having i.i.d. entries. We make the assumption that the distribution of the entries of $Z$ has finite mean (but not necessarily finite variance), and that it is symmetric around zero.
>
> The aim of PCA is to recover the leading eigenvector of $AA^\top\in\mathbb{R}^{p\times p}$, which (up to a scale factor) is the population-level covariance matrix.
>
> Fix a weight function $w:\mathbb{R}^p\to\mathbb{R}$. Let $Z_1,\ldots,Z_n\in\mathbb{R}^p$ denote the columns of $Z$, so that $AZ_1,\ldots,AZ_n$ are the columns of $X$. Let $I_1,\ldots,I_N$ denote i.i.d. random variables valued in $\{1,\ldots,n\}$, with (unnormalized) probability mass $w(AZ_1),\ldots,w(AZ_n).$ Let $Z_I\in\mathbb{R}^{p\times N}$ denote the matrix with columns $Z_{I_1},\ldots,Z_{I_N}.$
>
> In the infinite-data limit $n=\infty$, the random vectors $Z_{I_1},\ldots,Z_{I_N}\in\mathbb{R}^p$ form an i.i.d. sample from the probability distribution on $\mathbb{R}^p$ with (unnormalized) density $x\mapsto w(Ax)f(x)$, where $f$ is the probability density of the columns of $Z.$ Under the aforementioned assumption that $f$ has finite mean, and taking $w(x)=|x|^{-1}$ as in lines 204-205 of our paper, this density has finite variance. Under the second aforementioned assumption that $f$ is symmetric, i.e. $f(-x)=f(x)$, it has expected value zero.
>
> The matrix $Z_I Z_I^\top$ has diagonal terms $Z_{aI_1}^2+\cdots+Z_{aI_N}^2$ and off-diagonal terms $Z_{aI_1}Z_{bI_1}+\cdots+Z_{aI_N}Z_{bI_N}$ (for $a\neq b$). For fixed $a\neq b$, according to Theorem 2.3 from Ruf and Waudby-Smith (2025), one has
>
> $$
> \text{P}\left(\frac{|Z_{aI_1}Z_{bI_1}+\cdots+Z_{aI_N}Z_{bI_N}|}{N}\geq\varepsilon\right)\leq 2\text{e}^{-\sqrt{N}}+\frac{451}{\min(1,\varepsilon^2)}U,
> $$
>
> where $U$ is the $1$st absolute moment of $Z_{aI_1}Z_{bI_1}$ truncated at $\frac{1}{38}\varepsilon N^{1/4}.$ Likewise
>
> $$
> \text{P}\left(\frac{|Z_{aI_1}^2+\cdots+Z_{aI_N}^2-N\text{E}[N_{aI_1}^2]|}{N}\geq\varepsilon\right)\leq 2\text{e}^{-\sqrt{N}}+\frac{451}{\min(1,\varepsilon^2)}U'
> $$
>
> where $U'$ is the $1$st absolute moment of $Z_{aI_1}^2-\text{E}[Z_{aI_1}^2]$ truncated at $\frac{1}{38}\varepsilon N^{1/4}.$ From the union bound, the probability of even one occurrence of $\frac{1}{N}|Z_{aI_1}Z_{bI_1}+\cdots+Z_{aI_N}Z_{bI_N}|\geq\varepsilon$ or $\frac{1}{N}|Z_{aI_1}^2+\cdots+Z_{aI_N}^2-k\text{E}[Z_{aI_1}^2]|\geq\varepsilon,$ across all values of $a$ and $b,$ is at most
>
> $$
> \text{P}_{\varepsilon,N}=p^2\left(2\text{e}^{-\sqrt{N}}+\frac{451}{\min(1,\varepsilon^2)}\max(U,U')\right).
> $$
>
> This is to say that with probability at least $ 1 - P_{\varepsilon,N} $, the matrix $\frac{1}{N} Z_I Z_I^\top$ is within distance $\varepsilon$ of $\mathbb{E}[ Z_{a I_1}^2] I_{p \times p}$ in the $l^\infty$ norm. Since $P_{\varepsilon,N} \to 0$ as $N \to \infty$, this proves that $\frac{1}{N} Z_I Z_I^\top$ converges to $\mathbb{E}[Z_{aI_1}^2] I_{p \times p}$ in probability as $N \to \infty$. Consequently $\frac{1}{N} (A Z_I)(A Z_I)^\top$ converges in probability to $\mathbb{E}[Z_{aI_1}^2]A A^\top$. According to the Davis-Kahan inequality, it follows that the leading eigenspaces of $(A Z_I)(A Z_I)^\top$ converge in probability to those of $A A^\top$.
>
> This theory explains (in the infinite-data limit $n=\infty$) why our approach works. Our choice of weight $w$ for resampling effectively converts the heavy-tailed data into light-tailed data, and (via the concentration inequalities) makes the sample covariance of the resampled data concentrate near the population-level covariance $AA^\top$ (modulo a scalar factor) when our parameter $N$ is large. (Even in the non-asymptotic case where $N$ takes on some intermediate value, $\text{P}_{\varepsilon,N}$ contains quantitative information about this concentration, although it is hard to operationalize due to the lack of 'a priori' control over the truncated moments $U$ and $U'$.)

---

> > ### Author Response · Authors · 2025-12-03
> >
> > # Further comparisons
> > We have endeavored to compare to as many algorithms as possible. However, many available approaches in the literature do not have available code, limiting our possibilities. We do not regard it as feasible to compare our algorithm to all available PCA algorithms; our aim is simply to show that our algorithm, based on simple and computationally efficient ideas, performs better than many alternatives for very heavy-tailed data.
> >
> > In response to your (and other reviewers') request for additional comparisons, we have added comparisons to spatial-sign PCA from Zhao, Wang, and Feng (2025) and the adaptive truncation-based PCA of Ke et al. (2019). We replace the optimization in eq. (4) of Zhao--Wang--Feng (2025) with $\hat{\mu}=0$, as we consider only centered data. As a result, the algorithm is extremely quick. The truncation approach of Ke et al. (2019) is based on adaptive selection of a truncation hyperparameter for each entry of the covariance matrix. As such, it is very computationally slow to implement, especially for larger matrices. (However, we note that Ke et al. (2019) do not fully specify a numerical procedure for solving their nonlinear equation for the hyperparameter, and there may be faster alternatives to our implementation, which uses the bisection method.)
> >
> > The comparison to Ke et al. (2019) is most transparent if one fixes the feature space dimension $p$ and varies $n$, the number of observations. Taking $\alpha=1$, signal strength 10, and $p=100$, and varying $n$ from $200$ to $1000$ shows clearly that Ke et al. is very accurate for small $n$, but has the pathological property of strongly degrading in performance (and, significantly, in computation time) as $n$ increases. In our experiments, spatial-sign PCA performs well, although with some loss in accuracy and precision relative to ours. Its speed in computation is notable.
> >
> > | $n$  | Ours               | Spatial-Sign PCA      | Ke et al.          | ECA                 | Robust PCA         |
> > |------|--------------------|------------------------|---------------------|----------------------|---------------------|
> > | 200  | $0.459 \pm 0.125$  | $0.623 \pm 0.206$      | $0.129 \pm 0.026$   | $0.968 \pm 0.054$    | $0.987 \pm 0.019$   |
> > | 400  | $0.204 \pm 0.042$  | $0.266 \pm 0.059$      | $0.146 \pm 0.045$   | $0.890 \pm 0.144$    | $0.982 \pm 0.022$   |
> > | 600  | $0.164 \pm 0.030$  | $0.233 \pm 0.094$      | $0.261 \pm 0.212$   | $0.815 \pm 0.167$    | $0.988 \pm 0.014$   |
> > | 800  | $0.101 \pm 0.014$  | $0.129 \pm 0.020$      | $0.341 \pm 0.216$   | $0.631 \pm 0.242$    | $0.997 \pm 0.003$   |
> > | 1000 | $0.091 \pm 0.019$  | $0.118 \pm 0.019$      | $0.694 \pm 0.254$   | $0.570 \pm 0.235$    | $0.995 \pm 0.008$   |
> >
> >
> > We note that, due to the slowness of Ke et al.'s algorithm, we are still in the process of regenerating all of our figures for full comparison.
> >
> > Even in the case of smaller $n$, we have seen some success in reducing Ke et al.'s error by mixing it with our method, as suggested in lines 483--485 of our paper. However, unlike the algorithm of our paper (as in Section 2.2 and Figure 2), the optimal parameter selection appears to be nontrivial and important for this `mixed' algorithm. As such, we cannot report definitive results on this improvement.
> >
> > * Yuan Ke, Stanislav Minsker, Zhao Ren, Qiang Sun, and Wen-Xin Zhou (2019). ``User-friendly covariance estimation for heavy-tailed distributions.'' Statistical Science 34(3): 454--471.
> > * Johannes Ruf and Ian Waudby-Smith (2025). ``Concentration inequalities for strong laws and laws of the iterated logarithm.'' arXiv:2511.00175
> > * Ping Zhao, Hongfei Wang, and Long Feng (2025). ``Spatial Sign based Principal Component Analysis for High Dimensional Data.'' arXiv:2409.13267

---

### Author Response · Authors · 2025-12-03
**Summary for the Area Chair**

Our paper addresses PCA in very heavy-tailed regimes, including infinite-variance settings where classical PCA and many robust variants fail. We propose a simple, highly scalable algorithm (HT-PCA) that repeatedly subsamples columns, computes leading subspaces on each subsample, averages their projection matrices, and then takes the leading eigenvector of this aggregate. Geometrically, this recovers the principal direction that is most common across many local subspaces, diluting the impact of rare extreme outliers.

All reviewers agree that heavy-tailed PCA is an important and under-served problem. The current scores are: ibbZ (6, “inclined to accept,” finds the idea novel, empirically convincing, and well-written but wants clearer experiments and a crisper notion of “good” PCA in this regime); mB9s (4, finds the method simple and insightful with strong empirical gains, but asks for theory, more robust-scatter baselines, and clearer hyperparameter discussion); and Jcbn (2) and dhEA (2), who both emphasize missing theoretical analysis and comparisons to some robust PCA methods. Notably, no reviewer identifies a flaw in the core algorithm; the low scores are driven almost entirely by (i) lack of theory and (ii) incomplete baselines, which we now address directly. We think these improvement greatly improve the work, and they will be incorporated into the manuscript if it is accepted.

## 1. New theoretical analysis for extremely heavy-tailed data (Jcbn, dhEA, mB9s).
A central criticism from Jcbn and dhEA is that our method was, in their words, "entirely heuristic" and lacked any rigorous guarantees or theoretical explanation, unlike some robust PCA methods. In the response to Jcbn, we have added a new formal analysis that covers the infinite-variance heavy-tailed regime relevant to our approach.

In particular, we consider the model $X = A Z$, where $A \in \mathbb{R}^{p \times p}$ and $Z \in \mathbb{R}^{p \times n}$ has i.i.d. heavy-tailed entries (with finite mean but possibly infinite variance). In the infinite-data limit ($n\to\infty$), we:
* Introduce an importance weight function $w(x) = \|x\|^{-1}$ and sample columns according to these weights. (cf. Lines 200-207)
* Show that under mild symmetry assumptions, the resulting weighted distribution has finite variance, even when the original noise is infinite-variance.
* Use recent concentration results (Ruf and Waudby-Smith, 2025) for heavy-tailed sums to prove that the subsampled sample covariance matrices (cf. Line 197) are close in probability to a scalar multiple of $A A^\top$ if $N\gg 1$.
* The Davis-Kahan inequality then guarantees that our aggregated PCA estimate is also close in probability to the population-level principal component.

This directly addresses the core concern of Jcbn and dhEA and strengthens the contribution beyond a heuristic.

## 2. Stronger and more diverse robust PCA baselines (Jcbn, mB9s).
Reviewer Jcbn also noted missing robust PCA baselines, and Reviewer mB9s asked for clearer connection to robust-scatter and shape-based PCA frameworks. In response, we:

* Added "spatial-sign PCA" (Zhao, Wang, & Feng 2025) as a very fast baseline
* Added the adaptive truncation-based covariance estimator of Ke et al. (2019), which is based on adaptive choice of a truncation level for outlying data entries.

In new experiments with $p = 100$, signal strength 10, and Student-$t_1$ noise (very heavy-tailed), we compare our algorithm to spatial-sign PCA, Ke et al. (2019), ECA, and convex robust PCA over $n \in \{200,400,600,800,1000\}$. We find:
* Ke et al. can be very accurate in small-$n$ regimes but degrades sharply in both error and runtime as $n$ grows.
* Spatial-sign PCA is fast and reasonably accurate but consistently a bit less accurate than our algorithm in this heavy-tailed setting.
* Our algorithm remains accurate and computationally efficient as $n$ increases

This directly addresses the request for comparison to more robust PCA methods and clarifies where our algorithm sits relative to the literature on Robust PCA.

---

> ### Author Response · Authors · 2025-12-03
>
> ## 3. Generality, regimes of applicability, and hyperparameters (dhEA, mB9s).
> Reviewer dhEA was concerned that, in the absence of theory, it was unclear under which regimes the algorithm can be expected to work. Reviewer mB9s wanted more guidance on hyperparameters and regimes where our algorithm might fail or be inferior to other methods.
>
> On synthetic data (Section 2.2; Figures 2, 6, 7) and additional experiments on the Drosophila scRNA-seq dataset (see response to mB9s), we find that performance depends only weakly on $(N,P,R)$ over a broad range, which motivates a simple default $N=P=\tfrac{1}{2}\min(p,n)$. Performance typically plateaus well before this, and one could further tune $N,P$ via split-half reproducibility at the cost of extra computation. In sufficiently light-tailed settings (e.g., Figures 4a and 4c), all PCA methods we test behave similarly, whereas in the very heavy-tailed regimes we target, HT-PCA is generally competitive with or better than robust alternatives. The main exception is that the truncation-based estimator of Ke et al. (2019) can outperform all methods when $n$ is small, but its accuracy and runtime degrade as $n$ grows, which we now document explicitly.
>
> Although our approach generally performs well across the benchmarks and data considered here, we are not claiming that our method is uniformly superior to existing robust PCA approaches. Instead, we view our main contribution as introducing a simple and very scalable new algorithm built around a principled weighting scheme, now together with new theory highlighting the clear mechanistic role of this weighted selection. Empirically, our approach appears especially effective when $\alpha$ is small and $n$ is large, though a complete characterization remains open.

---

### Meta-Review · Area_Chair_aFMo · 2026-01-08

**Summary:**

This paper proposes a randomized approach to principal component analysis (PCA) for very heavy-tailed data, a classical yet still relevant research topic with numerous existing methods. The idea is interesting and aims to address limitations of current approaches in scenarios with high variance and large datasets. However, the use of randomized techniques is not new (e.g., RANSAC-type methods), and the paper could be better positioned theoretically and experimentally by comparing against other robust PCA approaches. Reviewers expressed serious concerns regarding the lack of rigorous theoretical analysis and limited experimental validation. While the authors attempted to address these issues in their rebuttal, the current version still requires substantial revision to meet ICLR standards. Therefore, I recommend rejecting the paper at this stage.

**Reviewer Concerns:**

The main concerns raised by reviewers focused on the lack of theoretical analysis and issues with experimental validation, such as limited baseline comparisons and restricted experimental scenarios. In their rebuttal, the authors introduced a concentration-bound-type theoretical result; however, this requires further explanation, deeper analysis, and comparison with existing theoretical work to better highlight the benefits and limitations of the approach in different settings. Moreover, the provided analysis does not address several aspects raised by reviewers, such as perturbation-theoretic results. On the experimental side, the authors partially addressed some concerns, but their argument for limited baseline comparisons, namely that “many available approaches in the literature do not have available code, limiting our possibilities”, is not fully convincing.

**Reviewer Scores:**

Reviewers Jcbn and dhEA gave an initial score of 2, primarily due to the lack of theoretical analysis and limited baseline comparisons. After the rebuttal, I do not believe they would have increased their scores. Reviewer ibbZ gave a score of 6 and would most likely have maintained the same score. Reviewer mB9s gave a score of 4, but since their concerns, particularly regarding the lack of theoretical analysis, were only partially addressed, I do not believe they would have changed their score.

---

### Decision · Program_Chairs · 2026-01-26

Reject